

# Genetic diversity of the breeding collection of tomato varieties in Kazakhstan assessed using SSR, SCAR and CAPS markers

Alexandr Pozharskiy[1,2,*], Valeriya Kostyukova[1,2,*], Marina Khusnitdinova[1], Kamila Adilbayeva[1,2], Gulnaz Nizamdinova[1], Anastasiya Kapytina[1], Nazym Kerimbek[1], Aisha Taskuzhina[1], Mariya Kolchenko[1], Aisha Abdrakhmanova[1], Nina Kisselyova[3], Ruslan Kalendar[4,5] and Dilyara Gritsenko[1]

[1] Laboratory of Molecular Biology, Institute of Plant Biology and Biotechnology, Almaty, Kazakhstan
[2] Department of Molecular Biology and Genetics, Al Farabi Kazakh National University, Almaty, Kazakhstan
[3] Fruit and Vegetable Research Institute, Almaty, Kazakhstan
[4] Helsinki Institute of Life Science HiLIFE, University of Helsinki, Helsinki, Finland
[5] National Laboratory Astana, Nazarbayev University, Astana, Kazakhstan
[*] These authors contributed equally to this work.

Corresponding author
Dilyara Gritsenko,
d.kopytina@gmail.com

## ABSTRACT

Tomato is one of the most prominent crops in global horticulture and an important vegetable crop in Kazakhstan. The lack of data on the genetic background of local varieties limits the development of tomato breeding in the country. This study aimed to perform an initial evaluation of the breeding collection of tomato varieties from the point of view of their genetic structure and pathogen resistance using a set of PCR based molecular markers, including 13 SSR markers for genetic structure analysis, and 14 SCAR and CAPS markers associated with resistance to five pathogens: three viruses, fungus *Fusarium oxysporum,* and oomycete *P hytophthora infestans.* Nine SSR markers were with a PIC value varying from 0.0562 (low information content) to 0.629 (high information content). A weak genetic structure was revealed in the samples of varieties including local cultivars and, predominantly, varieties from Russia and other ex-USSR countries. The local varieties were closely related to several groups of cultivars of Russian origin. Screening for a set of resistance markers revealed the common occurrence of the resistance locus *I* against *Fusarium oxysporum* and only the occasional presence of resistance alleles of other markers. No markers of resistance to the three considered viruses were revealed in local tomato varieties. Only two local cultivars had markers of resistance to *P. infestans,* and only the 'Meruert' cultivar had a combination of resistance markers against *P. infestans* and *F. oxysporum.* The obtained results have demonstrated the need for further studies of local tomato varieties with a wider range of molecular markers and source germplasm to lay a foundation for the development of tomato breeding in Kazakhstan.

# INTRODUCTION

Tomato (*Solanum lycopersicum* L.) is a representative plant species of the Solanaceae family, which includes a number of important vegetable and technical crops. Tomato is one of the most popular vegetable crops all over the world, as well as the closely related species, potato (*Solanum tuberosum* L.) (*Camargo Filho & Camargo, 2017*).

Tomatoes comprise an important part of overall vegetable production in Kazakhstan, with 788,760 tons harvested from 30.2 thousand hectares in 2022. Tomato production has been developed in the country extensively rather than intensively; the growing area has doubled, but the yield per hectare volume has stagnated in the last 30 years (*Food Agriculture Organization of the United Nations, 2021*). Among the tomato varieties approved for cultivation in the country, foreign cultivars prevail with a significant share of varieties from Russia and other ex-USSR countries (*The Ministry of Agriculture of the Republic of Kazakhstan, 2009*). Such a dependence on imported planting material poses various risks for food security, the most concerning of which is the possible importation of dangerous pests (*Chalam et al., 2021*), weeds (*Wilson et al., 2016*), and pathogens (*Elmer, 2001*; *Rodoni, 2009*). Thus, it is important for the domestic market of agricultural crops to adopt a wider use of old and newly obtained varieties that are bred locally, and it should be associated with comprehensive plant epidemiological controls. To confront potentially deleterious plant pathogens, it is not only necessary to detect and eradicate infected plants in a timely manner, but also to increase the resistance potential of cultivated crops against disease by breeding and selecting varieties with genetic factors of resistance. Modern practices require the extensive utilization of molecular methods to solve both these problems. Molecular markers associated with disease resistance in plants play a crucial role in modern breeding programs since their use in marker-assisted selection (MAS) helps to significantly reduce the time and labor required for developing new resistant varieties (*Collard & Mackill, 2008*; *Miedaner, 2016*). Such an approach utilizes molecular markers with known linkage with the target traits to lead selection without the need for direct control of the phenotype, *e.g.*, in the early developmental stages; the practices of MAS are widely utilized in tomato breeding for resistance to pathogens (*Foolad & Panthee, 2012*). However, in Kazakhstan, the implementation of such advanced breeding practices for tomato is limited by relatively low economic and scientific interests. To date, no systematic efforts have been made to lay the molecular genetic basis for selection programs for tomato crops. In contrast, the molecular genetics of wheat, the crop playing a prominent role in both the country's domestic food marker and international trade, has received significant research attention for years (*Kokhmetova et al., 2017*; *Anuarbek, Abugalieva & Turuspekov, 2018*).

The objective of this work was to investigate the genetic structure of the collection of tomato varieties deposited in the Fruit and Vegetable Research Institute (Almaty, Kazakhstan). The collection included established local cultivars along with varieties from abroad, predominantly from Russia and other ex-USSR countries. Most of them have not been included in the state register of crop varieties recommended for use (*The Ministry of Agriculture of the Republic of Kazakhstan, 2009*) and thus require extensive investigations of such factors as their genetic compositions, immunity, and physiological features under

local growth conditions. Along with previously published data on the genetic markers of resistance against three common viruses (*Pozharskiy et al., 2022*), this work presents the results of the first molecular genetic study of tomato varieties in Kazakhstan. A set of simple sequence repeats (SSRs), sequence characterized amplified region (SCARs), and cleavage amplified polymorphic sequences (CAPS) markers was used to evaluate the relations between selected cultivars and identify varieties bearing known loci of resistance to common tomato pathogens: oomycete *Phytophthora infestans,* fungus *Fusarium oxysporum,* tomato mosaic virus (ToMV), tomato spotted wilt virus (TSWV), and tomato yellow curly leaf virus (TYLCV). Except for *F. oxysporum,* these pathogens have been included in the list of quarantine objects, invasive species, and dangerous organisms by the Ministry of Agriculture of the Republic of Kazakhstan (*The Ministry of Agriculture of the Republic of Kazakhstan, 2015*). Three viruses, ToMV, TSMV, and TYLCV, are among the most dangerous tomato pathogens causing significant damage, potentially as much as the total yield loss (*Broadbent, 1976*; *Pico', Jo Diez & Nuez, 1996*; *Roselló, Díez & Nuez, 1996*). The broad specificity of these viruses to diverse host plant species (*Ying & Davis, 2000*; *Parrella et al., 2003*; *Hancinský et al., 2020*) expands the potential risks of virus propagation beyond tomato culture and makes disease control more challenging. Although, because of the lack of systematic molecular studies of tomato viruses, the presence of these viruses has not been detected to date in Kazakhstan, they are considered potentially threatening quarantine objects, as mentioned above. Previously, we tested a selection of tomato varieties using a set of SCAR and CAPS markers associated with resistance to the three mentioned viruses (*Pozharskiy et al., 2022*): PrRuG86-151, associated with resistance locus *Tm-2* against ToMV (*Lanfermeijer, Warmink & Hille, 2005*; markers NCSw-003, NCSw-005, NCSw-011, NCSw-012 (*Panthee & Ibrahem, 2013*), and Sw5-2 (*Dianese et al., 2010*), associated with resistant locus *Sw-5* against TSWV; markers Ty2-UpInDel, Ty3-InDel, Ty3-SNP9, and Ty3-SNP17, associated with resistance locy *Ty-2* and *Ty-3* against TYLCV (*Kim et al., 2020*). Here, we tested these markers on additional tomato samples from the local collection.

Oomycetes of the *Phytophthora* genus are among the most destructive plant pathogens, and *P. infestans* is the most threatening pathogen of potato and tomato, potentially causing total yield losses at the regional level (*Legard, 1995*; *Judelson & Blanco, 2005*; *Ismailova et al., 2017*; *Nowicki et al., 2012*; *Jung et al., 2015*). In Kazakhstan, *P. infestans* is among the most common tomato infections caused by fungus-like organisms (*Ismailova et al., 2017*). Due to the high genetic variability of this pathogen, the known resistance loci in tomato have only limited protective effect specific to particular *Phytophthora* isolates (*Nowicki et al., 2012*). The CAPS markers used here, TG328 and Ph3-gsm, are linked with the *Ph-3* resistance locus (*Robbins et al., 2010*; *Wang et al., 2016*), which confers partial resistance to a range of Phytophthora isolates and is widely used in breeding practices (*Jung et al., 2015*).

*Fusarium oxysporum* is a soil fungus capable of causing an opportunistic infection in a wide range of susceptible plants, including tomato; the hyphae of the fungus can penetrate the roots and colonize xylem vessels, causing vascular wilt (*Pietro et al., 2003*). The sub-species *F. oxysporum* f.sp. *lycopersici* (*Fol*) is the main causative agent of vascular wilt in tomato; three races are known, and for each of them the corresponding genetic factors of resistance have been described (*Chitwood-Brown et al., 2021*). The presence of
multiple *F. oxysporum* f.sp. *lycopersici isolates* has been detected in Kazakhstan (*Sagitov, El-Habbaa & El-Fiki, 2010*). Here, we tested our collection using dominant SCAR markers *At2* and *Z1063*, associated with resistance loci *I* and *I-2* (*Arens et al., 2010*), conferring resistance to races *Fol-1* and *Fol-2* (*Chitwood-Brown et al., 2021*).

This work aimed to fill the existing knowledge gap in the genetic basis of tomato breeding and Kazakhstan, to test the applicability of known genetic markers to local tomato varieties, and to identify genotypes bearing resistance markers against several important pathogens. As no studies of the genetic diversity of tomato have been lead to date in Kazakhstan, the obtained results will provide novel data on the state of tomato breeding in the country and help lay a basis for an initial inventory of tomato plant materials to be used both in agriculture and in breeding programs in Kazakhstan.

## MATERIALS AND METHODS

A selection of tomato varieties was obtained from the collection of the Fruit and Vegetable Research Institute (FVRI; Almaty, Kazakhstan) (Table 1). Seed materials were grown and DNA was isolated as previously described in *Pozharskiy et al. (2022)*.

SSR genotyping was conducted using known markers (Table 2) (*Smulders et al., 1997*; *Areshchenkova & Ganal, 2002*). Forward primers labeled with either fluorescein (FAM) or hexachlorofluorescein (HEX) were used for all markers. The polymerase chain reaction (PCR) conditions were set in accordance with the corresponding published protocols. The presence of PCR products was confirmed by electrophoresis in 1% agarose gel with 1x tris-acetate buffer, and then the fragment sizes (alleles) were determined by capillary electrophoresis using a 3500 Genetic Analyzer (Applied Biosystems, Thermo Fisher Scientific, Waltham, MA, USA). PCR samples were 20-fold diluted and combined into groups for multiplex fragment reading. Three groups were defined based on the used primer labels and expected fragment size ranges of the markers, to avoid overlaps between markers and to ensure the independent detection of alleles. The diluted PCR mixes were added to high-purity formamide (1 µl PCR mix, 0.15 µl LIZ(-500) Size Standard (Applied Biosystems, Thermo Fisher Scientific, Waltham, MA, USA), 8.85 µl formamide), denatured at 95 °C for 4 min, cooled on ice for 5 min, and loaded for capillary electrophoresis. Genotypes were determined using GeneMapper software and analyzed using a Bayesian approach implemented in MrBayes (*Ronquist et al., 2012*) and STRUCTURE (*Pritchard, Stephens & Donnelly, 2000*) software. R language (*R Core Team, 2019*) with the additional packages, indicated below, was used for general data handling and visualization. The genotyping data were encoded using an additive pseudo-haploid scheme where each observed allele was represented as a single digit value: 0 for absence, 1 for heterozygous state, and 2 for homozygous state. Minor allele frequency, and expected and observed heterozygosity for each marker were calculated using the 'adegenet' R package (*Jombart, 2008*; *Jombart & Ahmed, 2011*). The polymorphism information content (PIC) was calculated using the method (*Botstein et al., 1980*) implemented in the 'polysat' R package (*Clark & Jasieniuk, 2011*).

MrBayes was run for 50,000,000 generations with the Dirichlet distribution model for standard data; every 2,000th generation was sampled and used for diagnostics by the average

Pozharskiy et al. (2023), *PeerJ*, DOI 10.7717/peerj.15683

**Table 1  List of studied tomato varieties.**

| Sample ID | Variety name | | Country of origin | Included to the State Register | Sample ID | Variety name | | Country of origin | Included to the State Register |
|---|---|---|---|---|---|---|---|---|---|
| T001 | Choportula* | | Georgia | | T290 | Gribnoye Lukoshko | | Russia | |
| T003 | Zagadka Prirody* | [Enigma of Nature]*,*** | Russia | | T292 | Sladkoyezhka | [Sweet-tooth] | Kazakhstan | |
| T005 | Idillia* | [Idyll] | Russia | | T296 | Super Exotic | | Russia | |
| T007 | Yablochnyi* | [Apple-like] | Uzbekistan | | T314 | Ranniy310* | [Early 310] | Belarus | |
| T008 | Shalun* | [Varmint] | Russia | | T316 | Yarkiy Rumyanets* | [Bright Blush] | Russia-Kazakhstan** | |
| T010 | Uragan* | [Hurricane] | Serbia | | T317 | N7952691322* | | Russia | |
| T012 | Semka* | [Seed] | Russia | | T319 | Malinovyi Slon* | [Crimson Elephant] | Russia | |
| T013 | Pavlina* | | Russia | | T320 | Palmira* | | Russia | |
| T016 | Pozhar* | [Fire] | Belarus | | T322 | Lambrusko* | | Russia | |
| T018 | Rassvet* | [Sunrise] | Kazakhstan | + | T325 | Principe Borghese | | Italy | |
| T019 | Denar* | | Netherlands | | T328 | Tolstushka* | [Fatty] | Russia | |
| T020 | Hybrid16155 | | Kazakhstan | | T330 | Local with Carrot- Leaf | | Kazakhstan | |
| T022 | Korolek | [Kinglet] | Russia | | T333 | Rassvet362* | [Sunrise 362] | Russia | + |
| T024 | Spiridon* | | Russia | | T335 | Anait | | Armenia | |
| T025 | Venera* | [Venus] | Kazakhstan | | T336 | 33 Bogatyrya | [33 Heroes] | Russia | |
| T026 | Grapefruit | | Russia | | T338 | Yablochnyi* | [Apple-like] | Kazakhstan | |
| T053 | Yusupovskiy | | Uzbekistan | | T340 | Magnat | | Russia | |
| T078 | Lipen* | | Ukraine | | T341 | Malvina | | Russia | |
| T114 | Zhiraf* | [Giraffe] | Russia | | T343 | Tuzlovets | | Russia | |
| T122 | Dama* | [Dame] | Ukraine | | T444 | Krasnaya Presnya | [Red Presnya] | Russia | |
| T150 | Heart-likeRed | | Kazakhstan | | T466 | Pobeditel | [Winner] | Russia | |
| T170 | Zhirik | | Russia | | T479 | Malets | [Small Boy] | Russia | |
| T185 | Malika* | | Russia | | T496 | Nicola* | | Russia | |
| T187 | Ruzha* | | Belarus | | T512 | Russian Delicacy | | Russia | |
| T194 | Kozyr* | [Trump] | Russia | | T539 | Gloria | | Moldova | |
| T211 | Sunnik* | | Armenia | | T562 | Mechta | [Dream] | Kazakhstan | |
| T217 | Costoluto Biorentino* | | Italy | | T595 | Meruert | | Kazakhstan | + |
| T221 | Monach* | [Monk] | Russia | | T606 | Novichok | [Newcomer] | Russia | + |
| T235 | Pyatnitca | [Friday] | Russia-Kazakhstan** | | T609 | Vostorg | [Delight] | Kazakhstan | + |
| T237 | Barmaley | | Russia | | T612 | Luchezarnyi | [Shiny] | Kazakhstan | + |

Pozharskiy et al. (2023), *PeerJ*, DOI 10.7717/peerj.15683

**Table 1** (*continued*)

| Sample ID | Variety name | | Country of origin | Included to the State Register | Sample ID | Variety name | | Country of origin | Included to the State Register |
|---|---|---|---|---|---|---|---|---|---|
| T247 | Kolokola Rossii | [Russian Bells] | Russia | | T625 | Samaladay | | Kazakhstan | + |
| T257 | Ayan | | Kazakhstan | | T628 | Yantarnyi | [Amber] | Kazakhstan | |
| T262 | Orange-Violet | | Kazakhstan | | T631 | Leader | | Kazakhstan | + |
| T266 | Lilliput Hybrid F1 | | Italy | | T634 | Samaladay[***] | | Kazakhstan | + |

**Notes.**

[*]Data on resistance markers against ToMV, TSWV, TYCLV taken from *Pozharskiy et al. (2022)*.

[**]Local breeding line based on Russian cultivars.

[***]Intermediate breeding line.

[****]Translations of the Russian names of cultivars.

**Table 2  Tomato SSR markers used for genotyping.**

| Marker name | PCR primers | Repeating pattern[*] | Expected allele range | Multiplex group | Source |
|---|---|---|---|---|---|
| LE20592 | F: 5′- FAM -CTGTTTACTTCAAGAAGGCTG<br>R: 5′-ACTTTAACTTTATTATTGCCACG | $(TAT)_{15-1}(TGT)_4$ | 165–172 | 1 | |
| LE21085 | F: 5′- FAM -CATTTTATCATTTATTTGTGTCTTG<br>R: 5′-ACAAAAAAAGGTGACGATACA | $(TA)_2(TAT)_{9-1}$ | 103–119 | 1 | |
| LELE25 | F: 5′- FAM -TTCTTCCGTATGAGTGAGT<br>R: 5′-CTCTATTACTTATTATTATCG | $(TA)_{13-1}$ | 222–225 | 2 | |
| LELEUZIP | F: 5′- HEX -GGTGATAATTTGGGAGGTTAC<br>R: 5′-CGTAACAGGATGTGCTATAGG | $(AAG)_{6-1}TT$ | 101-105 | 2 | |
| LEMDDNA | F: 5′- HEX -ATTCAAGGAACTTTTAGCTCC<br>R: 5′-TGCATTAAGGTTCATAAATGA | $(TA)_9$ | 210-226 | 3 | |
| LEPRP4 | F: 5′- HEX -TTCATTTCTTGCAACTACGAT<br>R: 5′-CATACTAGCAACATCAAAGGG | $(TAT)_3(TGT)_5$ | 108-112 | 3 | |
| LESODB | F: 5′- FAM -TTATCAATTCATCATTGTGGC<br>R: 5′-AGTAAGGGGTTTAGGGGTAGT | $(TTC)_6$ | 208–212 | 1 | |
| LEATRACAb | F: 5′- FAM -GTATGTCAAATCTCTCTTGCG<br>R: 5′-ACTCTCCATCGTCTCTTTCAC | $(GA)_7$ | 184–186 | 2 | *Smulders et al. (1997)* |
| LPHSF24 | F: 5′- HEX -TTGGATTTACAAGTTCGATGT<br>R: 5′-GCATTTGACTTGATAGCAGTC | $(TA)_6$ | 156–158 | 1 | |
| LECHSOD | F: 5′- FAM -TTATCAATTCATCATTGTGGC<br>R: 5′-AGGGGTAGTGACAGCATAAAG | $(CTT)_6$ | 196–198 | 3 | |
| LEMDDNb | F: 5′- FAM -TAAATACAAAAGCAGGAGTCG<br>R: 5′-GAGTTGACAGATCCTTCAATG | $(TG)_4(TA)_4$ | 278–280 | 2 | |
| TMS63 | F: 5′- HEX -GCAGGTACGCACGCATATAT<br>R: 5′-GCTCCGTCAGGAATTCTCTC | $(AT)_4(GT)_{18}(AT)_9$ | 130–150[**] | 2 | *Areshchenkova & Ganal (2002)* |
| TMS58 | F: 5′- HEX -CATTTGTTGTATGGCATCGC<br>R: 5′-CAGTGACCTCTCGCACAAAA | $(TA)_{15}(TG)_{17}$ | 223–226[**] | 3 | |

**Notes.**

[*] (-1) at the subscript indicates the presence of an imperfect repeat.

[**] According to *Mazzucato et al. (2008)*; otherwise according to *Castellana et al. (2020)*.

standard deviation of tree probabilities in two parallel runs. The parameters of the run were monitored using built-in MrBayes statistics and Tracer (*Rambaut et al., 2018*), and the summary tree was generated using a burn-in threshold of 50%. The 'ggtree' R package (*Yu et al., 2017*) was used to visualize the summary tree along with the data mentioned below.

The STRUCTURE analysis was run for expected numbers of clusters $K$ from 1 to 10 using the standard admixture model with 50,000 burn-in and 100,000 Markov chain Monte-Carlo (MCMC) iterations. To find the optimal $K$, ten replicates were calculated for each $K$ value, and the CLUMPAK web-server (*Kopelman et al., 2015*) was used to estimate $\Delta K$ following Evanno's method (*Evanno, Regnaut & Goudet, 2005*).

PCR was performed for previously known markers of resistance against pathogens in accordance with published protocols (Table 3). All PCR products were checked using agarose gel electrophoresis. Markers requiring restriction (CAPS) were digested

by corresponding enzymes in a 20 µl mix containing 5 µl of the PCR mix, 0.5 µl of enzymes, and 2 µl of the appropriate restriction buffer, according to the manufacturer's recommendations. Restriction was performed overnight with the regular enzyme or for an hour with the enzymes of the FastDigest™ product series (Thermo Fisher Scientific, Waltham, MA, USA). The results of the restriction were evaluated by electrophoresis in 1.5% agarose gel with 1x tris-acetate buffer. All results of the genotyping by resistance markers were interpreted in accordance with the results reported in the source publications. For 31 specimens, previously published data on ToMV, TSWV, and TYCLV resistance markers were used for comparison (*Pozharskiy et al., 2022*), as indicated in Table 1.

PCR conditions for all markers used in the study are shown in File S1.

For all individual PCR reactions, both for SSR and resistance markers, the samples failing to produce a result were re-processed at least twice. If no results were obtained in any replicate, the genotype was reported as missing.

## RESULTS

A total of 68 tomato varieties were used in this study, including 13 cultivars of domestic origin. Most of these varieties represent a pool of tomato genotypes used in ongoing breeding programs. The local cultivars 'Meruert', 'Vostorg', 'Luchezarnyi', and 'Samaladay', as well as the Russian cultivars 'Novichok' and 'Rassvet 362', have also been approved for commercial use in Kazakhstan (*The Ministry of Agriculture of the Republic of Kazakhstan, 2009*).

According to the results of SSR genotyping, four markers—LEPRP4, LESODB, LECHSOD, and LEMDDNb—were revealed to be monomorphic across all tomato varieties (Table 4). LEPRP4 also had the highest missing genotype rate among all markers (11.76%). Markers LELE25, LELEUZIP, and LECHSOD were amplified in all studied samples. None of the other markers exceeded a missing rate of 7.35%, corresponding to 5 of 68 missing samples. Among the polymorphic markers, LEATRACAb, LPHSF24, and TMS58 had levels of observed heterozygosity that did not significantly differ from the expected values. The LEMDDNA marker had a slightly higher observed heterozygosity (*p*- value 0.0003; significance level 0.001); the other five markers had significantly lower observed values compared to the expected values (*p*- values near zero). Considering the nature of the studied samples, which comprised a heterogeneous set of specimens of different varieties rather than a single population, we did not expect the samples to follow Hardy–Weinberg equilibrium, and thus deviations between the expected and observed levels of heterozygosity were not surprising. Although the volume and heterogeneity of the samples limited any possible genetic inferences of the population, it could be speculated that the LEATRACAb, LPHSF24, and TMS58 markers were neutral with respect to the selection of tomato varieties. Markers LELEUZIP and LEMDDNA were revealed to be the most informative for the genotype discrimination, based on calculated PIC values 0.5328 and 0.629, respectively. Markers LEATRACAb and LPHSF24, in contrast, had low PIC values, 0.0570 and 0.0562, respectively. Five other polymorphic markers had moderate information content, with PIC values varying from 0.2035 (LELE25) to 0.3253 (TMS63).

**Table 3  Tomato SCAR and CAPS markers associated with resistance to pathogens.**

| Pathogen | Resistance locus | Linked marker | PCR primers | Restriction enzyme | Source |
|---|---|---|---|---|---|
| *Phytophtora infestans* | *Ph-3* | CAPS Ph3.gsm | F: 5′-TAGTATGGTCAAACATATGCAG R: 5′-CTTCAAGTTGCAGAAAGCTATC | FD *Hin* cII | *Wang et al. (2016)* |
| | | CAPS TG328 | F: 5′-GGTGATCTGCTTATAGACTTGGG R: 5′-AAGGTCTAAAGAAGGCTGGTGC | FD *Mva* I (*Bst* NI)*** | *Robbins et al. (2010)* |
| *Fusarium oxysporum* | *I* | SCAR At2 | F: 5′-CGAATCTGTATATTACATCCGTCGT R: 5′-GGTGAATACCGATCATAGTCGAG + control (LAT): F: 5′-AGACCACGAGAACGATATTTGC R: 5′-TTCTTGCCTTTTCATATCCAGACA | – | |
| | *I2* | SCAR Z1063 | F: 5′-ATTTGAAAGCGTGGTATTGC R: 5′-CTTAAACTCACCATTAAATC + control (Rubisco): F: 5′-ATGTCACCACAAACAGAGAC R: 5′-CTCACAAGCAGCAGCTAG | – | *Arens et al. (2010)* |
| Tomato mosaic virus (ToMV) | *Tm2* | CAPS PrRuG086-151 | F: 5′-GAGTTCTTCCGTTCAAATCCTAAGCTT GAGAAG R: 5′-CTACTACACTCACGTTGCTGTGATGCAC | *Ksp* AI (*Hpa* I)*** | *Lanfermeijer, Warmink & Hille (2005)* |
| | | SCAR NCSw-003 | F: 5′-TCTCGTTATCCAATTTCACC R: 5′-GCAATTTTGTTTCTTGGTCT | – | |
| | | SCAR NCSw-012 | F: 5′-ATGGTCAACTCGATCAGAAC R: 5′-TTTGGTGAGGATCTGATTTC | – | |
| Tomato spotted wilt virus (TSWV) | *Sw-5* | CAPS NCSw-007 | F: 5′-GTTGCTAACTCGACTCGTTC R: 5′-TCACTCACGTCCTATTGACA | FD *Hin* fI | *Panthee & Ibrahem (2013)* |
| | | CAPS NCSw-011 | F: 5′-TATCATCCTCATACCCCTTG R: 5′-GGATTTTCTCATCATCTCCA | *Hpy* F3I (*Dde* I)*** | |
| | | SCAR Sw5-2 | F: 5′-AATTAGGTTCTTGAAGCCCATCT R: 5′-TTCCGCATCAGCCAATAGTGT | – | *Collard & Mackill (2008)* |
| | *Ty-2* | SCAR Ty2-UpInDel | F: 5′-ACCCCAAAAACATTTCTGAAATCCT R: 5′-TGGCTATTTTGTGAAAATTCTCACT | – | |
| Tomato yellow curly leaf virus (TYLCV) | *Ty-3* | CAPS Ty3-InDel/SNP9 | F: 5′-CCTATCCTCAGTGTTTCGGTCA R: 5′-GGCGAAAGACTTTGTGTACACA | *Bst* 1107I (*Bst* Z17I) / *Mun* I (*Mfe* I)*** | *Kim et al. (2020)* |
| | | CAPS Ty3-SNP17 | F: 5′-TCTCAGGTGATGCTGAGCAC R: 5′-AGAGAACGAAAACGAAATTTCAAACA | *Rsa* I | |

**Notes.**
*Gene ID and genomic positions according *S. lycopersicum* genome assembly SL3.0.
**Marker positions in *S. lycopersicum* genome assembly SL3.0.
***Isoschizomers used in the work and by the original authors (in parentheses).
FD, FastDigest™ restriction enzyme product series (Thermo Fisher Scientific, USA).

The genetic heterogeneity of the studied samples was revealed by Bayesian cluster analysis (Figs. 1A, 1B). The results obtained using two algorithms implemented in MrBayes and STRUCTURE software were compared to acquire a more detailed picture of the genetic structure of the samples. According to the MrBayes results, most of the studied tomato varieties formed a large subtree with a weak sub-structure. The results obtained with STRUCTURE produced a data partition into five clusters, in accordance with the best Evanno's ΔK value (Fig. 1D). The first cluster (shown cyan) was the most distinct group

**Table 4  Summary of SSR genotyping of 68 tomato varieties.**

| Marker name | $N$ | Detected alleles | Missing genotype rate | MAF | $H_e$ | $H_o$ | $H_e$ vs. $H_o$ ($\chi^2$ test $p$- value) | PIC |
|---|---|---|---|---|---|---|---|---|
| LE20592 | 3 | 164,167,170 | 0.0147 | 0.1045 | 0.3206 | 0.0149 | 0 | 0.2972 |
| LE21085 | 2 | 103,117 | 0.0441 | 0.1769 | 0.2912 | 0.0154 | $2.2315 \times 10^{-14}$ | 0.2488 |
| LELE25 | 3 | 218,220,222 | 0 | 0.0735 | 0.3334 | 0.2059 | $6.7279 \times 10^{-14}$ | 0.2035 |
| LELEUZIP | 4 | 102,104,105,106 | 0 | 0.3088 | 0.5978 | 0 | 0 | 0.5328 |
| LEMDDNA | 5 | 211,213,219,227,233 | 0.0147 | 0.2463 | 0.6788 | 0.7164 | 0.0003 | 0.6290 |
| LEPRP4 | 1 | 201 | 0.1176 | – | – | – | – | – |
| LESODB | 1 | 207 | 0.0294 | – | – | – | – | – |
| LEATRACAb | 2 | 184,186 | 0.0294 | 0.0303 | 0.0588 | 0.0606 | 0.7995 | 0.0570 |
| LPHSF24 | 2 | 158,164 | 0.0147 | 0.0298 | 0.0579 | 0.0597 | 0.8011 | 0.0562 |
| LECHSOD | 1 | 195 | 0 | – | – | – | – | – |
| LEMDDNb | 1 | 277 | 0.0147 | – | – | – | – | – |
| TMS63 | 4 | 158,184,188,202 | 0.0735 | 0.2222 | 0.3818 | 0.0793 | 0 | 0.3253 |
| TMS58 | 3 | 226,228,230 | 0.0735 | 0.1667 | 0.3287 | 0.3333 | 0.8085 | 0.2916 |

**Notes.**

$N$, number of detected alleles; MAF, minor allele frequency; $H_e$, expected heterozygosity; $H_o$, observed heterozygosity; PIC, polymorphism information content.

representing a compact sub-group at the tree; the highest probabilities were assigned to the 'Lipen' (Ukraine), 'Yablochnyi [Apple-like]' (Uzbekistan), 'Choportula' (Georgia), and 'Shalun [Varmint]' (Russia) cultivars, which had identical genotypes. The local variant of the 'Yablochnyi [Apple-like]' cultivar was the only variety from Kazakhstan included in this cluster; however, it was located apart from other varieties in the tree and differed from its Uzbekistani relatives in two markers, LE21085 and TMS58. Another distinct cluster (shown in yellow) included two small subclusters in the tree; the typical members of this group were the 'Ayan' (Kazakhstan), 'Ruzha' (Belarus), 'Nicola' (Russia), and 'Pyatnica [Friday]' (local breeding line based on Russian cultivar) cultivars. The other three clusters (shown in red, blue, and purple) appeared as a mixed set of subgroups and intermediate genotypes within the main subtree.

Fifteen tomato varieties resulted from breeding efforts established in Kazakhstan. All local varieties yielded a high genetic similarity according to the SSR markers used (Fig. 1E; File S2). Across all 11 polymorphic markers, only three markers demonstrated genotype variations within the local cultivars: LEMDDNA with detected alleles 211, 213, 227, and 233; LELEUZIP with alleles 102, 105, and 106; and TMS58 with alleles 226, 228, and 230. The LELE25, LEATRACAb, and TM63 markers had only two differing genotypes across 15 local varieties, and marker LE20592 had the only differing genotype in the 'Sladkoyezhka' cultivar. This cultivar was the most distinct across all local varieties. The 'Yantar [Amber]', 'Leader', 'Luchezarnyi [Shiny]', 'Meruert', 'Vostorg [Delight]', and 'Mechta [Dream]' varieties formed a group of similar genotypes (purple color in Fig. 1B), along with Russian cultivars 'Novichok [Newcomer]', 'Korolek [Kinglet]', 'Rassvet 365 [Sunrise 365]', and '33 bogatyrya [33 heroes]'. The breeding line of the 'Samaladay' cultivar (specimen T634) also belonged to this group; however, the finally established line for commercial use (specimen T625) differed in the LELEUZIP (genotype 102/102) and

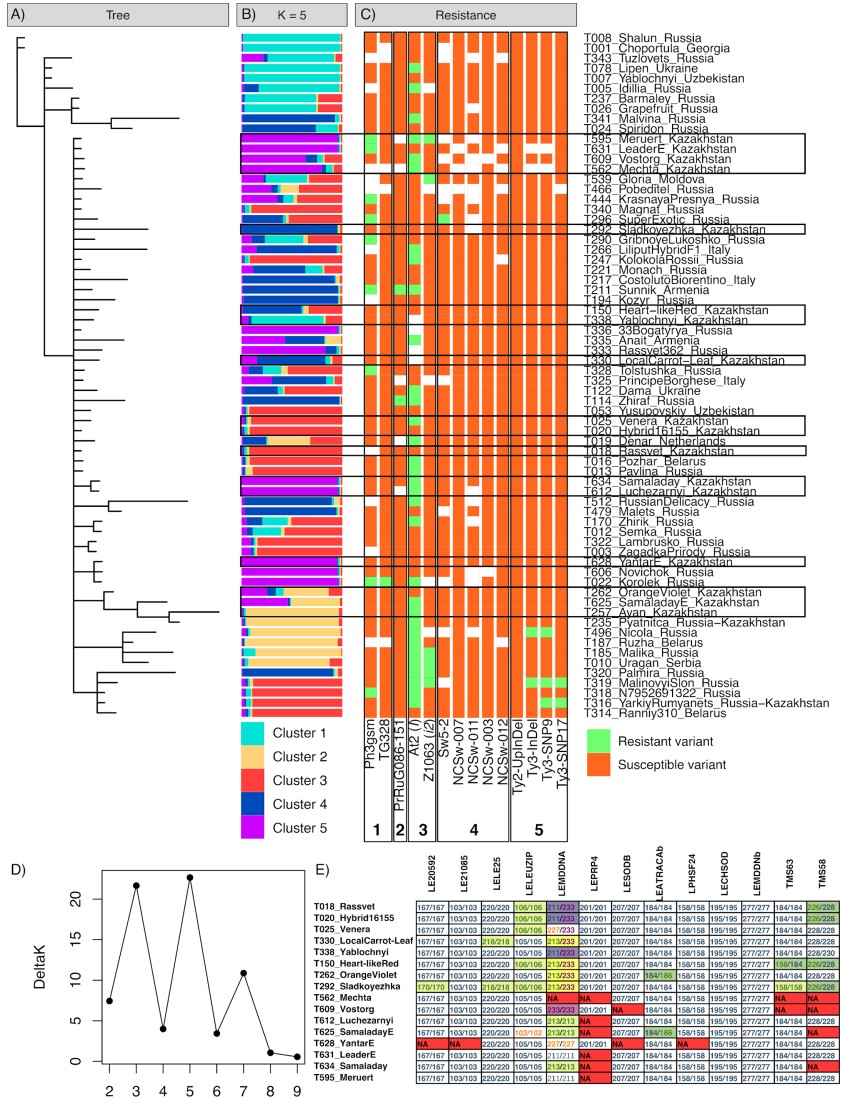

**Figure 1** **Results of the genotyping of tomato varieties with SSR markers and markers associated with disease resistance.** (A) Bayesian tree of varieties based on SSR markers. (B) STRUCTURE plot for five cluster configurations based on SSR markers. (C) tomato genotypes in markers of resistance against *Phytophtora infestans* (1), ToMV (2), *Fusarium oxysporum* (3), TSWV (4), and TYCLV (5). (D) Evanno's ∆*K* plot indicating the optimal *K*. (E) variations of SSR genotypes in tomato varieties of Kazakhstani origin.

LEATRACAb (184/186) markers. The LEPRP4 and TMS58 markers were characterized by a notably high occurrence of missing genotypes in this group. These local varieties were obtained by the breeding programs of the former Research Institute of Potato and Vegetable Breeding (now part of the Fruit and Vegetable Research Institute, Almaty, Kazakhstan) (*Kurganskaya & Dzhantasova, 2005*). Other local varieties were more diverse, in relation to various foreign cultivars.

The analysis of SCAR and CAPS markers associated with resistance against infections revealed the prevailing presence of resistance loci to fungus *Fusarium oxysporum* and

**Table 5** Summary of the genotyping results of 68 tomato varieties with SCAR and CAPS markers of resistance against infectious diseases.

| Pathogen | Marker | Marker type | Susceptible genotypes | | Resistant genotypes | | Missing data counts |
|---|---|---|---|---|---|---|---|
| | | | Fragment sizes[**] | Counts | Fragment sizes[**] | Counts | |
| *Phytophtora infestans* | Ph3.gsm | CAPS | $596 + 501 + 107$ | 48 | $596 + 291 + 258$ | 9 | 11 |
| | TG328 | CAPS | 500 | 62 | 260+240 | 1 | 5 |
| *Fusarium oxysporum* | At2 | SCAR | 92 | 32 | 130+92 | 32 | 4 |
| | Z1063 | SCAR | 1380 | 57 | 1380+940 | 6 | 5 |
| Tomato mosaic virus (ToMV) | PrRuG086-151[*] | CAPS | NA[***] | 61 | NA[***] | 2 | 5 |
| Tomato spotted wilt virus (TSWV) | NCSw-003[*] | SCAR | 600 | 66 | 680 | 0 | 2 |
| | NCSw-012[*] | SCAR | 1000 | 62 | – | 0 | 6 |
| | NCSw-007[*] | CAPS | 240 | 65 | 480 | 0 | 3 |
| | NCSw-011[*] | CAPS | 600 | 53 | 430+200 | 0 | 15 |
| | Sw5-2[*] | SCAR | 510 or 464 | 56 | 574 | 1 | 11 |
| Tomato yellow curly leaf virus (TYLCV) | Ty2-UpInDel[*] | SCAR | 213 | 68 | 120 | 0 | 0 |
| | Ty3-InDel[*] | CAPS | 669 | 64 | 353+325 | 2 | 2 |
| | Ty3-SNP9[*] | CAPS | 555+114 | 63 | 678 | 3 | 2 |
| | Ty3-SNP17[*] | CAPS | $562 + 148 + 52 + 51$ | 65 | $497 + 148 + 65 + 52 + 51$ | 2 | 1 |

**Notes.**
[*] Including data from *Pozharskiy et al. (2022)*, as indicated in Table 1.
[**] According corresponding publications, see references in Table 3.
[***] Fragment sizes were not reported by *Lanfermeijer, Warmink & Hille (2005)*. The genotypes were accessed based on the reference gel image from the referenced article.

oomycete *Phytophtora infestans,* compared to viruses (Table 5, Figs. 1C; File S3). The most commonly occurring marker was At2, associated with resistance locus *I* against *F. oxysporum*; half of all 64 successfully genotyped samples were positive for resistance. Another resistance marker against *F. oxysporum*, Z1063, associated with *I2* resistance genes, was observed in six specimens, including the local 'Meruert' cultivar. Both markers are dominant SCAR markers linked with the corresponding resistance loci introduced to tomatoes from *Solanum pimpinellifolium (Arens et al., 2010)*. Two codominant markers, Ph3-gsm and TG328, have been linked with the *Ph-3* locus conferring resistance to *P. infestans (Robbins et al., 2010; Wang et al., 2016)*. Two local cultivars, 'Meruert' and 'Leader', had the resistant allele of Ph3-gsm; the only specimen with the resistant variant of TG328 was the Russian cultivar 'Korolek [Kinglet]'. Only two cultivars had the resistant allele of marker PrRuG086-151 associated with locus *Tm-2* conferring resistance to ToMV (*Lanfermeijer, Warmink & Hille, 2005*), Russian cultivar 'Zhiraf [Giraffe]' and Armenian 'Sunnik', as was previously revealed by *Pozharskiy et al. (2022)*. Almost no markers associated with the resistant locus *Sw-5* against TSWV (*Dianese et al., 2010; Kim et al., 2020*) were detected, with the exception of marker Sw5-2 in the Russian 'Super exotic' variety. For TYCLV, markers associated with resistance loci *Ty-2* and *Ty-3* were tested (*Kim et al., 2020*). No resistant allele for the marker Ty2-UpInDel was revealed. Three markers associated with the resistant variant of *Ty-3* were previously identified in Russian cultivars (*Pozharskiy et al., 2022*).

## DISCUSSION

The results of this study reflect the history and current state of tomato breeding in Kazakhstan. Since the collapse of the Soviet Union in 1991, the development of vegetable breeding and seed production has remained stagnant in independent Kazakhstan due to a shortage of funding and highly qualified experts (*Amirov, 2012*). The results of the present study have revealed a low genetic diversity of local tomato varieties and their similarity to foreign cultivars. The content of the studied collection of varieties, as well as the list of approved cultivars (*The Ministry of Agriculture of the Republic of Kazakhstan, 2009*), show the predominant presence of tomato varieties of Russian origin. Such dependence on Russian seed material, which could be traced back to the Soviet period, not only makes local horticulture more vulnerable to political and economic factors, but also decreases the diversity of the genetic resources available for cultivation.

The set of SSR markers used in this study showed limited information content when applied to the considered collection of tomato varieties. According to *Botstein et al. (1980)*, PIC values above 0.5 indicate high information content of a codominant marker, values between 0.25 and 0.5—moderate information content, and values below 0.25— low information content. Of 13 markers used, only two were highly informative, three were moderately informative, four had PIC below 0.25, and four were monomorphic. Consequently, the genetic structure revealed by the Bayesian analysis was weak and provided little information on the possibly classification of the local varieties. Thus, to obtain a molecular genetic basis for tomato breeding in Kazakhstan, further studies are required, following two conditions: (a) a sufficient number of markers covering most parts of the tomato genome; and (b) a wider range of available tomato germplasm from throughout the world, or available data on their diversity and compatibility with used marker sets.

A set of SCAR and CAPS markers of resistance to five diseases revealed a low abundance of corresponding resistance factors not only in the local cultivars, but also in all those studied here. The most common marker, At2, associated with resistance locus *I* against *F. oxysporum,* had an equal proportion of resistant and susceptible variants across varieties; approximately the same ratio, 8:7, was observed in the group of local cultivars. However, this marker displayed no strong genotype distribution pattern in relation to the SSR data. Another *F. oxysporum* resistance marker, Z1063 (locus *I-2*), had an allele associated with resistance in one local cultivar, 'Meruert'. Based on the specificity of the associated resistance loci to *Fol* races (*Chitwood-Brown et al., 2021*), resistance to race *Fol-1* is more common than *Fol-2*; further studies should also test resistance factors against *Fol-3*. Four local cultivars had a resistant genotype in the Ph3-gsm marker to *P. infestans,* and no local varieties had resistance markers against the three considered viruses. These results indicate that no systematic approaches have been developed thus far to work with resistance factors in breeding; the observed markers appeared occasionally and without a strong relation to the overall genetic structure.

Despite the role of the former Research Institute of Potato and Vegetable Breeding, in general, the development of tomato breeding in Kazakhstan has been led in a poorly

organized and sporadic manner. Because of the losses of information resulting from outdated infrastructures and insufficient funding since the early years of the country's independence, the origin and subsequent selection of local tomato varieties cannot be traced. The re-establishment of tomato selection in the country at the contemporary level will require joined efforts from the government, farming businesses, and research institutions, including the utilization of modern methods of molecular genetics.

The obtained results demonstrate that further studies with expanded sets of markers and varieties are required, as the data obtained in this work provide limited information. The extension of knowledge about tomato genetics is a crucial aspect of the development of tomato breeding in the country, and particular attention should be paid to the evaluation of a wider range of markers associated with resistance to various diseases and other biotic and abiotic stress factors, supplementing experimental tests. The development of new resistant varieties and their introduction for wide-scale commercial usage will increase the sustainability of the tomato market in Kazakhstan and, thus, help strengthen food safety in the republic. Marker-assisted selection should therefore play a key role in breeding to achieve this goal.

## CONCLUSIONS

The results of this study demonstrated the low diversity and weak genetic structure of tomato varieties bred and used in Kazakhstan. The set of 13 SSR markers tested has shown limited applicability for studying the genetic structure of local tomato varieties. The local varieties have shown a low abundance of genetic markers associated with resistance to *Phytophthora infestans* and *Fusarium oxysporum,* and the absence of resistance markers against ToMV, TSMV and TYCWV. The limitations of the obtained results imply the need for further studies employing a wider range of markers and involving more diverse tomato genotypes, which are important for the future development of tomato breeding in Kazakhstan.

### Funding
The work was performed in the framework of a targeted funding program BR18574149 "Development of highly productive cultivars and lines of agricultural crops using innovative technologies" (Ministry of Science and Higher Education of the Republic of Kazakhstan). The funders had no role in study design, data collection and analysis, decision to publish, or preparation of the manuscript.

### Grant Disclosures
The following grant information was disclosed by the authors:
Development of highly productive cultivars and lines of agricultural crops using innovative technologies (Ministry of Science and Higher Education of the Republic of Kazakhstan, targeted funding program BR18574149).

## Competing Interests

Ruslan N. Kalendar is an Academic Editor for PeerJ.

## Author Contributions

- Alexandr Pozharskiy conceived and designed the experiments, analyzed the data, prepared figures and/or tables, authored or reviewed drafts of the article, and approved the final draft.
- Valeriya Kostyukova performed the experiments, analyzed the data, prepared figures and/or tables, authored or reviewed drafts of the article, and approved the final draft.
- Marina Khusnitdinova performed the experiments, authored or reviewed drafts of the article, and approved the final draft.
- Kamila Adilbayeva performed the experiments, authored or reviewed drafts of the article, and approved the final draft.
- Gulnaz Nizamdinova performed the experiments, authored or reviewed drafts of the article, and approved the final draft.
- Anastasiya Kapytina performed the experiments, authored or reviewed drafts of the article, and approved the final draft.
- Nazym Kerimbek performed the experiments, authored or reviewed drafts of the article, and approved the final draft.
- Aisha Taskuzhina performed the experiments, authored or reviewed drafts of the article, and approved the final draft.
- Mariya Kolchenko performed the experiments, authored or reviewed drafts of the article, and approved the final draft.
- Aisha Abdrakhmanova performed the experiments, authored or reviewed drafts of the article, and approved the final draft.
- Nina Kisselyova performed the experiments, prepared figures and/or tables, authored or reviewed drafts of the article, selected and managed the source plant material, and approved the final draft.
- Ruslan Kalendar conceived and designed the experiments, prepared figures and/or tables, authored or reviewed drafts of the article, and approved the final draft.
- Dilyara Gritsenko conceived and designed the experiments, prepared figures and/or tables, authored or reviewed drafts of the article, and approved the final draft.

## Data Availability

The raw data is available in the Supplemental Files.

## Supplemental Information

Supplemental information for this article can be found online at http://dx.doi.org/10.7717/peerj.15683#supplemental-information.

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
