# Peer review of "Genetic diversity of the breeding collection of tomato varieties in Kazakhstan assessed using SSR, SCAR and CAPS markers"

_PeerJ, doi:10.7717/peerj.15683_

## Round 0.1 · original submission · Major Revisions

Dear authors

Reviewers have elaborated on all the shortcomings that need to be addressed. Please review carefully and make any necessary corrections.

Reviewer 1 ·

Basic reporting

Dear authors
The following modifications are required:
Abstract
 In general, this section is not well worded. It is simply written. This section should contain the most important results obtained in this study. So, this section should be improved
 The authors must define, before the describing the aim, the issue clearly in a single line and explain why they chose this approach to study this research?
 The authors should provide some scored values such as MAF, H0, He, number of genotypes in terms of resistance and susceptible…. etc.
 No information about the genetic structure is available
 No data about the absence and presence of markers is available in this manuscript.
 The authors should present a decisive conclusion that is derived from the research in the final line of the abstract and provide a single line of future prospects.
Introduction
 L43-47: These sentences should be supported by the references
 Some information about the detrimental effects of pathogens infecting the tomato plant should be added in particular the pathogens named in this manuscript
 Some information about the type of markers used in this study should be included especially in the selection of resistance and susceptible genotypes
 Some information about the MAS should be included
 The authors should give some lines about the knowledge gap which their research has covered along with the hypothesis statement
 Also, the authors should provide a novelty statement at the end. What new things authors have done or correlated in this research compared to old ones?
Materials and Methods
 The reaction and program of PCR should be inserted.
 The concentration of agarose should be mentioned
 All abbreviations in Materials and methods should be given in their full name for the first time.
Results
 L143-147: These lines should be deleted because they are repeated in materials and methods
 L148-149: For supporting this result, it is better to put the agarose gel
 Why did not the authors measure the diversity indices, PIC and AMOVA analysis for SSR markers?
 The authors should explain the results of structure by showing the pure and admixture genotypes
 The size of allele or marker associated with the resistance should be included
Discussion
 This section is generally poorly written. Some sentences are not related to the interpretation of results and are written as a review, particularly in the SSR data information.
Conclusion
 The authors have written this section in a straightforward manner, and they should conclude the most important findings.

Experimental design

It is acceptable

Validity of the findings

The findings need some improvement

Annotated reviews are not available for download in order to protect the identity of reviewers who chose to remain anonymous.

Reviewer 2 ·

Basic reporting

Although it has minor shortcomings, it was still an important work with effort. All criticisms here are also included in the text.
L 3. What kinds of markers?
L 23: tomato breeding replicated 2 times.
L39-L40. There are a lot of keywords, please reduce them, and as many as possible keywords should be in the main title.
Line 45 and line 48, line 50. There is no citation in this passage.
Line 100. For the first using the long name should be given
L. 101. Marker informations, names, bp, repeating locus etc
L. 105. (%? Is the agarose gel),

Experimental design

Without polymorphic information (PIC) values how can possible to do genetic diversity analysis? Please indicate how to harvest and obtain the delta K value.

To obtain Tree what kind of software did you use? Which software did you use to obtain allele number, all frequency, heterozygosity, homozygosity, and so on… did you use Microsoft Excel, or PowerMarker software, or maybe another?
The methodology is wrong. First admixture model with 50,000 burn-in and 100,000 MCMC iterations, then K value and the number of iterations should be done in STRUCTURE software analysis.

Validity of the findings

In almost whole genetic diversity analyses geographical information pictures like a map that shows the location of varieties.

(Please add some gel images related to those markers, no matter monomorphic and polymorphic. We need to track and match the results and gel images.

Almost the whole discussion section is irrelevant. Results should be discussed in this section and comparing the literature, comments are made about the similarities and differences.

Normally genetic diversity is the first step of plant development by hybridization techniques, with genetic diversity we can determine the kinship levels of populations and As parents, we determine the genotypes to be included in the breeding program with genetic diversity analyzes. In the conclusion part, markers were mentioned. The information of the markers is already available. Do you need to test it again?

Which software did you use to obtain the phylogenetic tree (Dendogram)

Additional comments

What is the importance of PIC?

The last passage of the introduction is the objective of the study. But unrelated and extremely long. yes clearly stated that objective is analysis the genetic structure of tomato population but this should not be the real target. This is a step forward to reach the real goal.

Annotated reviews are not available for download in order to protect the identity of reviewers who chose to remain anonymous.

---

## Round 0.2 · accepted · Accept

Dear Authors,

I am pleased to inform you that your manuscript accepted for publication. I commend you for improving your manuscript immensely by paying the utmost attention to the reviewers' and editorial recommendations.

Congratulations and good luck with your next research.

Reviewer 2 ·

Basic reporting

You can publish. From my side it is ok.

Experimental design

You can publish. From my side it is ok.

Validity of the findings

You can publish. From my side it is ok.